



**Palaeo-environmental evolution of Central Asia during the Cenozoic: New insights from**
**the continental sedimentary archive of the Valley of Lakes (Mongolia)**
Andre Baldermann[1*], Oliver Wasser[1], Elshan Abdullayev[2,3], Stefano Bernasconi[4], Stefan
Löhr[5], Klaus Wemmer[6], Werner E. Piller[7], Maxim Rudmin[8] and Sylvain Richoz[9]
[1]Institute of Applied Geosciences, Graz University of Technology, NAWI Graz Geocenter, Graz,
Austria; baldermann@tugraz.at; oliver.wasser@student.tugraz.at
[2]Department of Life Sciences, Khazar University, Baku, Azerbaijan; elabdulla3@gmail.com
[3]Department of Geoscience, French-Azerbaijani University (UFAZ), Baku, Azerbaijan;
elabdulla3@gmail.com
[4]Geological Institute, ETH Zurich, Zurich, Switzerland; stefano.bernasconi@erdw.ethz.ch
[5]Department of Earth and Environmental Sciences, Macquarie University, Sydney, Australia;
stefan.loehr@mq.edu.au
[6]Geoscience Centre (GZG), University of Göttingen, Göttingen, Germany; kwemmer@gwdg.de
[7]Institute of Earth Sciences, University of Graz, NAWI Graz Geocenter, Graz, Austria;
werner.piller@uni-graz.at
[8]Division of Geology, Tomsk Polytechnic University, Tomsk, Russia; rudminma@tpu.ru
[9]Department of Geology, University of Lund, Lund, Sweden; sylvain.richoz@geol.lu.se
*Corresponding author: Andre Baldermann
Institute of Applied Geosciences, Graz University of Technology, 8010, Graz, Austria
Tel: +43 316 873 6850, Fax: +43 316 873 6876; E-mail: baldermann@tugraz.at
Key Words: Cenozoic climate change; Central Asia; Palaeo-environment; Westerly winds;
Hydroclimate; Paleosols



**Abstract**
The Valley of Lakes basin (Mongolia) contains a unique continental sedimentary archive,
suitable for constraining the influence of tectonics and climate change on the aridification of
Central Asia in the Cenozoic. We identify the sedimentary provenance, the (post)depositional
environment and the palaeo-climate based on sedimentological, petrographical, mineralogical
and (isotope) geochemical signatures recorded in authigenic and detrital silicates as well as soil
carbonates in a sedimentary succession spanning ~34 to 21 Ma. The depositional setting was
characterized by an ephemeral braided river system draining prograding alluvial fans, with
episodes of lake, playa or open steppe sedimentation. Metamorphics from the northern adjacent
Neoarchean to late Proterozoic hinterlands provided a continuous influx of silicate detritus to
the basin, as indicated by K-Ar ages of detrital muscovite (~798-728 Ma) and discrimination
function analysis. The authigenic clay fraction is dominated by illite-smectite and "hairy" illite
(K-Ar ages: ~34-25 Ma), which formed during coupled petrogenesis and precipitation from
hydrothermal fluids originating from major basalt flow events (~32-29 Ma and ~29-25 Ma).
Changes in hydroclimate are recorded in $\delta^{18}O$ and $\delta^{13}C$ profiles of soil carbonates and in silicate
mineral weathering patterns, indicating comparatively humid to semi-arid conditions prevailed
in the late(st) Eocene, changing into arid conditions in the Oligocene and back to humid to
semi-arid conditions in the early Miocene. Aridification steps are indicated at ~34-33 Ma, ~31
Ma, ~28 Ma and ~23 Ma and coincide with some episodes of high-latitude ice sheet expansion
inferred from marine deep-sea sedimentary records. This suggests long-term variations of the
ocean/atmosphere circulation patterns due to $p$CO$_2$ fall, re-configurations of ocean gateways
and ice-sheet expansion in Antarctica could have impacted the hydroclimate and weathering
regime in the basin. We conclude that the aridification in Central Asia was triggered by reduced
moisture influx by westerly winds driven by Cenozoic climate forcing and the exhumation of
the Tian Shan and Altai mountains and modulate by global climate events.



## 1. Introduction

The Cenozoic Era (66 Ma to the present day) saw several dramatic changes of the marine and continental ecosystems (e.g., evolution of large plankton feeders such as baleen whales, shift towards cold-water, high nutrient plankton assemblages at high latitude, expansion of terrestrial mammals) major tectonic events (e.g., opening of Southern Hemisphere Oceanic gateways, shift to the 4-layer structure of the modern ocean, collision of the African-Arabian-Eurasian plates, uplift of the Alpine and Himalayan mountain belt) and global climate forcing (e.g., change from greenhouse to icehouse conditions) (Cerling, 1997; Houben et al., 2013; Norris et al., 2013; Cermeño et al., 2015; Mutz et al., 2018; Komar and Zeebe, 2021). The acceleration of Cenozoic climate cooling started after the Early Eocene Climatic Optimum (EECO; ~52-50 Ma), with temperatures ~10-12 °C warmer than the modern deep ocean, followed by the appearance and expansion of the Antarctic ice-sheets after the Eocene-Oligocene Transition (EOT; ~34 Ma) and ultimately culminating in the extensive Northern Hemisphere glaciation of the Pleistocene (~2.6-0.01 Ma; Zachos et al., 2001; Lear et al., 2008; Mudelsee et al., 2014; Abdullayev et al., 2021). This long-term transition in Earth`s climate is well documented in marine sedimentary archives, but its impact on the evolution of continental ecosystems remains poorly constrained, mainly because continuous, well preserved terrestrial records are scarce and the responses to climate change in these settings are highly complex, depending on latitude, proximity to coast and mountain ranges, position relative to climatic winds, vegetation etc. (e.g., Caves Rugenstein and Chamberlain, 2018; Baldermann et al., 2020). An exception is the sedimentary archive of the Valley of Lakes (Mongolia), which hosts a ~34-21 Ma record of continental sedimentation in Central Asia. The biostratigraphy and the correlation between different outcrops in this basin are well established based on mammalian communities and gastropod records (Harzhauser et al., 2017), magnetostratigraphy (Sun and Windley, 2015) and radiometric age dating of different basalt horizons (Daxner-Höck et al., 2017), rendering this



locality suitable for constraining the links between tectonism and climate change in Central
Asia during the Cenozoic. The Eocene to Miocene of Central Asia was characterized by
accelerated aridification (Dupont-Nivet et al., 2007; Xiao et al., 2010; Bosboom et al., 2014;
Li et al., 2016), expressed as a substantially expanded Gobi Desert relative to today (Guo et
al., 2008; Lu et al., 2019) and a sudden turnover in the mammal record (Harzhauser et al., 2016;
Barbolini et al., 2020). Several, partially opposing hypotheses have been proposed to explain
the aridification of Central Asia, including a combination of orbitally-driven climate forcing,
the stepwise retreat of the proto-Paratethys Sea and uplift of the Tibetan Plateau (Pälike et al.,
2006; Zhongshi et al., 2007; Li et al., 2020) or a continuous decrease of moisture transport by
the westerlies due to exhumation of the Tian Shan and Altai mountains (Caves et al., 2014;
Caves et al., 2015; Caves Rugenstein and Chamberlain, 2018). However, the evolution of
Central Asia`s hydroclimate in the Cenozoic was not a period of continuous aridification;
indeed, the climatic conditions in particular in the Oligocene were highly complex and
characterized by numerous glacial-interglacial cycles (Xiao et al., 2012). Recently, Richoz et
al. (2017) have identified two aridification pulses in Central Asia, in the early and late
Oligocene, which they assigned to global climatic events. To date, a correlation of the global
marine record with the terrestrial record of Mongolia is barely developed (Harzhauser et al.,
2016; Harzhauser et al., 2017; Richoz et al., 2017), which limits our understanding of the
relative influences of climate change and regional tectonics on the evolution of hydroclimate
and weathering conditions in Central Asia in the Cenozoic.
In this contribution, we greatly extend the existing mineralogical and (isotope) geochemical
dataset previously reported in Richoz et al. (2017) for the Eocene-Miocene sediments from the
Valley of Lakes (Mongolia): K-Ar ages and polytype analysis of detrital and authigenic illitic
phases coupled with discrimination function analysis and sedimentological-petrographical-
geochemical inspection are used to constrain provenance, palaeo-environmental conditions and



post-depositional alteration history of this sedimentary succession. Systematic, coherent
changes in the weathering patterns of silicate detritus and pristine $\delta^{18}O$ and $\delta^{13}C$ signatures
recorded in paleosols carbonates allow us to revise and refine the evolution of hydroclimate
and weathering conditions in Central Asia in the Cenozoic.

**2. Geological framework**
The Valley of Lakes is an ESE-WNW striking sedimentary basin with ~500 km extension in
largest dimension. It is located in Central Mongolia and bordered by the Khangai mountains in
the north and the Gobi Altai mountains in the south (Fig. 1a). The geological super-units in the
north of Mongolia contain Neoarchean, Proterozoic and Palaeozoic rocks of the Caledonian
orogen as well as late Neoproterozoic to Ordovician (Tuva-Mongol) magmatic arc and related
back- and fore-arc intrusions, accretionary wedge sequences and ophiolites (Porter, 2016). The
geological super-units in the south are characterized mainly by a Palaeozoic orogen, especially
the Kazakh-Mongol magmatic arc, which forms the border between Mongolia and China.
These units include mainly Devonian to Carboniferous island arc volcanic rocks, Ordovician
to Silurian volcanics, Ordovician to Carboniferous metamorphosed sedimentary sequences and
Permo-Carboniferous granitoids (Porter, 2016).
Regarding the regional lithostratigraphic context, the northern structural units of the Valley of
Lakes basin in the Taastsiin Gol area comprise dominantly fault- and thrust-bounded crystalline
basement of Neoarchean to Palaeozoic age (Fig. 1b). These include the Baidrag (high-grade
gneisses, charnockites and amphibolites, up to 2.65 Ga old) and the Burdgol zone (metapelites,
metapsammites and metacherts, 699 ± 35 Ma) in its southernmost end (Teraoka et al., 1996).
Further structural units towards the north are the Bayan Khongor (metamorphosed basic rocks,
ophiolites and pelitic schists, 450 Ma), the Dzag (metapelites and metapsammites, 440 ± 22
Ma and 395 ± 20 Ma) and the Khangai zone (unmetamorphosed, but tectonically deformed



sandstones, mudstones and intercalated olistolith sequences of unspecified Devonian to
Carboniferous age) (Teraoka et al., 1996; Höck et al., 1999). All of these zones are intruded by
numerous granitoids of variable age (Proterozoic to Cretaceous) and composition (Höck et al.,
1999). The major zones located in the south of the Valley of Lakes basin comprise the Baga
Bogd, the Ikh Bogd and the Bogd som, which are petrographically indistinguishable from the
time-equivalent metasediments and metavolcanics of the Bayan Khongor zone and of the
Permian quartzitic conglomerates from the adjacent Mount Ushgoeg (Höck et al., 1999).
In the focus of this study are the fossiliferous siliciclastic sediments of the Taatsiin Gol Basin,
which record important information about changes in sediment provenance, weathering paths
and conditions and palaeo-climate in Central Asia during the Eocene to Miocene. The herein
investigated sedimentary sections span the Tsagaan Ovoo Formation (upper Eocene), the
Hsanda Gol Formation (Oligocene) and the Loh Formation (lower Miocene). Five sections,
namely Taatsiin Gol right (TGR-AB), Taatsiin Gol south (TGR-C), Hsanda Gol (SHG-D),
Tatal Gol (TAT-E) and Hotuliin Teeg (HTE), were chosen for this study, because of the well-
constrained biostratigraphy at these localities. These sections form an integrated sedimentary
succession with a thickness of ~115 m (Richoz et al., 2017). Two prominent stratigraphic
marker beds, the basalt I group (32.4-29.1 Ma) and the basalt II group (28.7-24.9 Ma) crop out
at ~40-41 m and at ~94-100 m in the sedimentary profile (Daxner-Höck et al., 2017). A younger
basalt III group (13.2–12.2 Ma) dates back to the middle Miocene, but is not part of the
sedimentary succession investigated here. Further details about the local nomenclature, the
investigated profiles, profile correlation and lithostratigraphic relationships are provided in
Harzhauser et al. (2017), Daxner-Höck et al. (2017) and Richoz et al. (2017). Due to the
complex architecture of the Valley of Lakes basin and adjacent areas, a mixed provenance has
been proposed for the basin fill, however, detailed knowledge about the palaeo-depositional
environment and source area relationships remain poorly constrained (Höck et al., 1999).



**3. Materials and Methods**
3.1 Materials
Representative bulk sediment samples (140 in total) were taken from different outcrops, which
cover the entire sedimentary succession of the Valley of Lakes from the upper Eocene to the
lower Miocene. The layers sampled vary in color, composition, texture, fossil and carbonate
content, etc., however, they do not show optical signs of alteration, such as recent surface
weathering. Samples for geochemical, isotopic and mineralogical analysis were crushed in a
ball mill for 10 min and micronized using a McCrone mill for 8 min, with ethanol addition.
Samples with a high clay mineral content based on an initial mineralogical inspection were
selected further for an identification of the clay mineral suite, which is defined here as $< 2\ \mu m$
size fraction (Rafiei et al., 2020). As for the clay mineral separation, 5 g of the bulk material
was reacted with 5 % HCl for 10 min to remove the carbonates, followed by standard Atterberg
sedimentation and subsequent collection and drying of the $< 2\ \mu m$ size fraction at 40 °C. Fast
acid digestion was used to reduce leaching or dissolution of the clay minerals under acidic
conditions (Baldermann et al., 2012). Four samples from the Hsanda Gol Formation with a
high amount of illitic phases were used for an illite polytype and K-Ar analysis. To this end,
three sub-fractions ($< 1\ \mu m$, 1-2 $\mu m$ and 2-10 $\mu m$) were separated by Atterberg sedimentation,
which all represent mixtures of authigenic illitic phases and detrital illite/muscovite.

3.2 Analytical methods
The major, minor and trace element composition of a sub-set of samples (91 in total) was
analyzed via a Philips PW2404 wavelength dispersive X-ray fluorescence (XRF) spectrometer.
Fine powdered samples (0.8 g) were heated to 1050 °C to remove the volatile components
($CO_2$, $H_2O$, etc.), following determination of the loss on ignition (LOI) by gravimetric analysis.
The residuals were fused at 1200 °C using $LiBO_2$ (4 g) as the fluent agent. The standard glass



tablets were analyzed together with a set of USGS standards (analytical error: ± 0.5 wt% for
the major elements; Richoz et al., 2017).
Sediment origin and variations in the detrital influx among the different provenance areas were
depicted using discrimination plots calculated on the basis of major oxide compositions (Roser
and Korsch, 1988). The weathering paths and intensities in the source rock areas were assessed
through changes in the weathering indices, such as the chemical index of alteration (CIA), the
chemical index of weathering (CIW) and the plagioclase index of alteration (PIA), which were
calculated based on the major oxide compositions using the following equations (Nesbitt and
Young, 1982; Abdullayev et al., 2021):
$CIA = (Al_2O_3 / (Al_2O_3 + CaO^* + Na_2O + K_2O)) \times 100$
$CIW = (Al_2O_3 / (Al_2O_3 + CaO^* + Na_2O)) \times 100$
$PIA = (Al_2O_3 - K_2O) / (Al_2O_3 + CaO^* + Na_2O - K_2O) \times 100$,
where $CaO^*$ denotes the fraction of CaO present in the silicate fraction. $CaO^*$ was calculated
by subtraction of the total CaO content of the bulk sediments (determined by XRF analyses)
from the CaO content associated with carbonate minerals (determined by XRD analyses, see
below). The weathering conditions of the source areas were identified further using $Al_2O_3$ –
$CaO^* + Na_2O - K_2O$ (A-CN-K) ternary diagrams (Nesbitt and Young, 1984).
The mineralogical composition of all bulk samples was determined by Rietveld-based analysis
of X-ray diffraction (XRD) patterns recorded on a PANalytical X'Pert PRO diffractometer
(Co-Kα; 40 kV and 40 mA) equipped with a high-speed Scientific X'Celerator detector. The
top loading technique was used for the preparation of randomly oriented samples, which were
examined in the range from 4-85 2θ with 0.008°2θ/s step size and 40 s count time. The
PANanalytical X`Pert Highscore Plus software and a pdf-4 database were used for mineral
quantification (analytical error: < 3 wt%; Baldermann et al., 2021). The separated grain size
sub-fractions were X-rayed under identical operational conditions. The amounts of authigenic



(1M and $1M_d$ polytype) and detrital ($2M_1$ polytype) illitic phases were calculated using the
following equations (Grathoff and Moore, 1996):
$\%2M_1 = 2.05 + 360 \times A_{(114)}/A_{(2.6\ \text{Å band})}$
$\%1M = 4.98 + 136 \times A_{(-112)}/A_{(2.6\ \text{Å band})}$
$\%1\,M_d = 100 - \%1M$ or $100 - \%2M_1$
where A is the area (in cps·2θ) of the polytype-specific hkl-reflections of illite and of the 2.6
Å band, respectively (analytical error: ~± 5 %; Baldermann et al., 2017).
Oriented clay films were prepared for the further characterization of the clay mineral fraction
(< 2 μm) using a Phillips PW 1830 diffractometer (Cu-Kα; 40 kV and 30 mA) outfitted with a
graphite monochromator and a scintillation counter. The clay films were prepared by mixing
50 mg of clay fraction with 5 mL of deionized water, following ultrasonic treatment in a water
bath for 10 min to produce a clay-in-suspension, which was subsequently sucked through a
porous ceramic tile of ~4 cm² size (Baldermann et al., 2014). The clay films were examined in
the range from 3-30° 2θ with 0.02° 2θ step size and 2 s/step count time, each at air-dry states,
after solvation with ethylene glycol (EG) and after heat treatment at 550 °C for 1 h. The
proportion of illite layers (%Ilt) in mixed-layered illite-smectite (Ilt-Smc) was calculated based
on the position of the 002-reflections obtained from XRD patterns of EG-solvated clay films
($d_{EG-002}$ in Å) following the equation (analytical precision: ± 5 %; Baldermann et al., 2017):
$\%\,Ilt = 60.8 \times d_{EG-002} - 504.5$.
Illite crystallization ages were calculated through coupled illite polytype and K-Ar analysis
carried out on the separated grain size sub-fractions. The $K_2O$ content of these samples was
determined in digested aliquots (1M HF and $HNO_3$ mixture) in duplicate via a BWB-XP flame
photometer™ using 1 % CsCl as the ionization buffer and 5 % LiCl as the internal standard.
The Ar isotopic composition was analyzed in a stainless steel extraction and purification line
connected to a Thermo Scientific ARGUS VI™ noble gas mass spectrometer operated in static



mode at the University of Göttingen (Germany). The radiogenic $^{40}$Ar content was measured
using the standard isotope dilution method applying a highly enriched $^{38}$Ar spike calibrated
against the biotite standard HD-B1. K-Ar age calculations were made based on the constants
recommended by the IUGS (for details see Wemmer et al., 2011). The grain size sub-fractions
are free of K-containing mineral phases other than mica/illite group minerals, which would
disturb the radiogenic K-Ar ages.
A scanning electron microscopy (SEM) study was carried out to characterize the mineralogy,
chemical composition, microfabrics and alteration patterns of the authigenic and detrital (clay)
minerals present in the sediments. Therefore, specimens were prepared on standard SEM stubs,
coated with carbon and analyzed using a GEMINI® Zeiss Ultra 55 microscope operated at 5-
15 kV of accelerating voltage and equipped with a high efficiency in-lens secondary electron
(SE) detector and an EDAX Si(Li)-detector for high-resolution imaging and energy-dispersive
X-ray spectrometry (EDX) analysis.
The $\delta^{13}$C and $\delta^{18}$O isotopic composition of the carbonate fraction was analyzed to constrain the
palaeo-climatic trends recorded in the paleosols. In a previous study (Richoz et al., 2017) it
was shown that the soil carbonates (calcrete nodules, lenses and crusts) mostly record pristine
$\delta^{13}$C and $\delta^{18}$O isotopic compositions reflective of conditions during their formation and are not
influenced by detrital or secondary carbonates, such as calcite spar or dolomite. The samples
(139 in total) were reacted with 102 % phosphoric acid at 70 °C in a Kiel II automated reaction
system and the liberated $CO_2$ gas analyzed with a ThermoFinnigan mass spectrometer MAT
Delta. The measured $\delta^{13}$C and $\delta^{18}$O values were corrected against the NBS19 standard and are
reported in per mill (‰) relative to the Vienna-PeeDee Belemnite (V-PDB) standard (analytical
precision: < 0.05 ‰ for $\delta^{13}$C and < 0.1 ‰ for $\delta^{18}$O; Richoz et al., 2017).


## 4. Results


4.1 Sediment petrography


An integrated lithostratigraphic profile of the investigated sedimentary succession (upper


Eocene to lower Miocene) from the Taatsiin Gol region, which is a part of the Valley of Lakes,


including the biozonation and some field impressions, is presented in Figure 2.


The sediments from the Tsagaan Ovoo Formation (upper Eocene) are dominantly coarse clastic


sand and gravel deposits of white-greyish color with embedded clay and silt layers of greyish-


yellow-green to reddish-brown color, depending on the Fe content (Richoz et al., 2017). The


coarser beds show cross-bedding and are frequently poorly sorted, while the finer layers show


trough and planar cross-bedding, lamination, inverse to normal grading, rarely ripples and


channel fills, and are better sorted. Roots and plant debris and bioturbation features, such as


burrows, indicate local paleosol formation (Richoz et al., 2017).


The overlying Hsanda Gol Formation (Oligocene) has a higher fossil content (mainly remains


of small mammals) and appears as horizontally bedded and poorly sorted clay to silt layers of


brick-red to reddish-brown color with intercalated cross-bedded sandstone beds and minor sand


and granule lenses of greyish color (Fig. 2c). Paleosol formation is documented by abundant


crypto- to microcrystalline calcite nodules and calcite crusts of centimeter to decimeter size


encapsulating soil and plant materials (Fig. 2b; Richoz et al., 2017). These calcrete layers of


greyish-white color are partially intergrown with Fe- and Mn-(oxy)hydroxides of orange-


greyish-black color. The basalt I and basalt II horizons are exposed at ~40-41 m and at ~94-


100 m and interfinger with the sediments from the Hsanda Gol Formation (Fig. 2b,d).


The Loh Formation (lower Miocene) comprises generally poorly sorted and structure-less silty-


clayey horizons with embedded pebbles and lenses of greyish-white to reddish-brown color as


well as trough to planar cross-bedded sand and gravel beds of greenish-yellow-red color, which


are deposited in alternate mode. Sedimentary structures seen in the coarser beds include inverse




to normal grading, ripple marks, channel and scour fills and overbank fines (Richoz et al.,
2017). Most horizons are highly fossiliferous (remains of small mammals and gastropods) and
show signs of paleosol formation, such as calcite nodules and crusts incorporating plant debris,
and burrow structures (Harzhauser et al., 2017).

4.2 Bulk and clay mineralogy
The mineralogical composition of the Valley of Lakes samples is dominated by quartz (10-55
wt%), illite/muscovite (10-50 wt%), calcite (0-70 wt%), feldspar (5-15 wt%; mainly albite and
plagioclase and minor orthoclase) and hematite (0-10 wt%) (Table S1). XRD analysis identifies
the illite/muscovite as an almost pure illitic phase composed of > 95 % Ilt layers and < 5 %
Smc layers (Fig. S1) with the $1M_d$ polytype structure dominating (~90-95 % of the total illite
fraction; Fig. S2). The proportions of the 1M and $2M_1$ polytype structures of illite do not exceed
~5-10 % of the total illite fraction. Kaolinite, chlorite (Mg-rich), mixed-layered Ilt-Smc
comprised of ~30-10 % Ilt layers and ~70-90 % Smc layers (Fig. S1) as well as Ti-oxides (rutile
and anatase) represent minor constituents (Fig. S2), accounting altogether for less than ~5 wt%
of the sediments. Trace amounts of zeolite and amphibole (< 5 wt%) are documented between
~35 and 45 m and between ~90 and 110 m in the sedimentary succession, i.e., adjacent to the
basalt I and II groups. Vermiculite, dolomite, ankerite, anhydrite, halite and pyrite were not
identified in the samples, which contrasts observations made by Höck et al. (1999).
The sediments from the Tsagaan Ovoo Formation have the highest proportions of quartz, illite,
feldspar and hematite and the lowest content of calcite compared to the other two formations,
consistent with less abundant calcrete horizons developed in the upper Eocene sediments (Fig.
3a). The sediments from the Oligocene Hsanda Gol and lower Miocene Loh formations have
highly variable, but on average higher calcite contents than the Tsagaan Ovoo Formation due
to abundant paleosol formation and related lower contents of silicate minerals and hematite



(Fig. 3b,c). The depletion of hematite in these samples argues for a detrital origin and for the
precipitation of this mineral phase on silicate detritus during sediment transportation under oxic
conditions. No systematic trends in the abundance of the mineral phases was observed across
the investigated profile (cf. Table S1).

4.3 Microfabrics and illite crystallization ages
A microstructural study of weakly consolidated samples taken from the Hsanda Gol Formation
reveals (sub)angular to rounded detrital quartz grains (Fig. S3a), which are partly overgrown
by diagenetic quartz cement (Fig. S3b), as well as partially dissolved feldspar grains (Fig. S3c).
Calichized areas are cemented by calcite spar, which appears as crypto- to microcrystalline
material with aggregate particle sizes in the micrometer to millimeter range (Fig. S3d). All
these components are covered or intergrown by fine hematite particles (Fig. S3e), although silt-
size hematite grains are also observable. Coarse chlorite flakes as well as tiny, rounded to
vermiform kaolinite particles are barely seen (Fig. S3f). Indeed, the clay mineral suite is
dominated by two types of illite and one type of Ilt-Smc. SEM-EDX analysis suggests the illites
have higher contents of $Al_2O_3$ and $K_2O$, but lower contents of $SiO_2$ and $Na_2O$ than the Ilt-Smc.
The illites occur either as micrometer-sized particles with platy or pseudohexagonal forms
being evenly dispersed throughout the matrix (type 1: Fig. 4a,c,e,g) or as long (micrometer-
scale), but thin laths and fibers, which grow into the open pore space (type 2: Fig. 4b,d,f,g,h).
The latter type of illite is often referred to as "hairy illite" (Güven et al., 1980; Rafiei et al.,
2020). The Ilt-Smc is a nanometer-sized material with flaky to irregular particle forms, which
covers detrital grains or grows into the open pore space (type 3: Fig. 4b,d,h).
When viewed together with the results of the illite polytype analysis and measured K-Ar ages
(Table 1), all sub-samples represent physical mixtures of detrital $2M_1$ illite/muscovite (type 1),
authigenic $1M_d$/1M illite (type 2) and authigenic $1M_d$ Ilt-Smc (type 3). Accordingly, the plot



of the proportion of $2M_1$ illite/muscovite against the K-Ar age of a given sub-sample (Fig. 5)
provides individual crystallization ages for the detrital and authigenic illitic phases (Grathoff
and Moore, 1996): The upper intercept of the best-fitting line at 100 % of $2M_1$ reveals the
crystallization age of detrital illite/muscovite, which is 727.6 to 797.9 Ma. The lower intercept
of the best-fitting lines at 100 % of $1M_d + 1M$ gives crystallization ages for the authigenic clay
minerals, which vary between 25.2 and 34.2 Ma.

4.4 Geochemistry and weathering indices
Variations in the major element composition of the samples (Table S2) follow changes in the
abundance of silicate minerals (e.g., quartz, feldspar and clay minerals) relative to calcite and
hematite across the sedimentary succession. No distinct trends among the different formations
are seen, except for a lower CaO content and higher contents of $SiO_2$, $Al_2O_3$, $K_2O$, $Na_2O$, MgO
and $Fe_2O_3$, on average, in the Tsagaan Ovoo Formation, compared to the Hsanda Gol and Loh
formations, corroborating the mineralogical and petrographic results (cf. Table S1 and Fig. 3).
Minor amounts of $TiO_2$ belong to rutile and anatase and traces of MnO and $P_2O_5$ correspond
to Mn-oxides and apatite. The positive correlations of Cu, Ga, Rb and Zn with $Al_2O_3$ as well
as Ce, La, Y and Zr with $TiO_2$ and Sr with CaO point to their association with clay minerals
(i.e., structural incorporation or sorption onto the clay mineral surface), heavy minerals and
carbonate minerals, respectively (Abdullayev et al., 2021). Ba, Co, Cr, Hf, Nb, Ni, Pb, Sc, Th,
V and U are inconspicuous due to lack of correlation with $Al_2O_3$ and $TiO_2$ or low concentration
in the samples.
The plot of the chemical data in the A-CN-K ternary diagram (Fig. 6) shows the samples fall
within or plot slightly above the compositional range of Post-Archean Australian Shale (PAAS)
and Average Proterozoic Shale (APS) and thus follow the predicted weathering trend for basalt
protoliths and Upper Continental Crust (UCC) rocks (Nesbitt and Young, 1984; Bahlburg and





Dobrzinski, 2011). The shift of most of the data toward the K pole of the diagrams indicates
K-metasomatism has affected the chemical composition of the sediments through the growth
of authigenic illite and Ilt-Smc (Fedo et al., 1995), consistent with petrographic observations
and clay polytype analyses. The CIA, CIW and PIA values vary from 70-83, 83-97 and 79-96
across the different formations, which averages of 79, 94 and 92 for the Loh Formation and 76,
90 and 88 for both the Hsanda Gol and Tsagaan Ovoo formations, respectively (Table S3).

4.5 Soil carbonate $\delta^{18}$O and $\delta^{13}$C isotopic composition
The $\delta^{18}$O and $\delta^{13}$C values of the soil carbonates vary in the range from -11.7 to -0.2 ‰ and -
8.1 to -3.8 ‰ across the sedimentary succession of the Valley of Lakes (Table S4). Six samples
taken close to the basalt I and II groups show comparatively lighter isotope values, -12.9 to -
8.6 ‰ of $\delta^{18}$O and -9.4 to -8.3 ‰ of $\delta^{13}$C, which indicates post-depositional overprinting.
Therefore, these samples are not considered further. A high scatter in $\delta^{18}$O values (-9.3 to -0.2
‰) and relatively light $\delta^{13}$C values (-7.5 to -6.4 ‰) are seen in the lower part of the Hsanda
Gol Formation, changing into less fluctuating $\delta^{18}$O values (-10.3 to -7.0 ‰) and systematically
heavier $\delta^{13}$C values (-7.6 to -3.8 ‰) in the middle and upper part of the Hsanda Gol Formation
until the lower Miocene. Around the series/stage boundary, a gradual shift towards lighter $\delta^{18}$O
values (-11.7 to -8.6 ‰) and fluctuating, but lighter $\delta^{13}$C values (-8.1 to -4.4 ‰) are evident.

**5. Discussion**
5.1 Sediment provenance
The time interval from the Neoarchean to the late Permian saw the development of large parts
of the fault- and thrust-bounded crystalline basement of Mongolia. The main lithological units
forming this basement include Neoarchean metamorphic rocks and Palaeozoic metasediments
and magmatic rocks, which are all intruded by volcanic and magmatic rocks of various age,





composition and provenance (Zorin et al., 1993). This complex architecture and the denudation
processes in the Mesozoic, which formed the Valley of Lakes basin and created the present-
day regional landscape and relief, are documented in the heavy mineral spectra of the Cenozoic
basin fill (Höck et al., 1999): the presence of epidote, amphibole, garnet, rutile, pyroxene,
sphene, zircon and tourmaline suggest that a mountainous region in the area of the present-day
Khangai mountains were the potential source areas (McLennan et al., 1993). Quartz, pegmatite,
granite, siltstone, basalt and carbonate clasts found in the gravel fraction (Höck et al., 1999)
are also indicative of a heterogeneous provenance for the Valley of Lakes sediments.
The major oxide compositions of the sediments from the Valley of Lakes mainly plot in the
"P4-quartzose sedimentary provenance" field and only a few samples plot into the "P1-mafic
igneous provenance" field in the Roser and Korsch (1988) discrimination diagram (Fig. 7).
This indicates metamorphosed sediments rich in quartz and poor in feldspar and subordinate
mafic to intermediate igneous and metamorphic rocks are the source rocks for the Valley of
Lakes sediments. These rock types are common to all lithological units exposed in the adjacent
lands of the Valley of Lakes (Höck et al., 1999). However, if considering the crystallization
ages of the $2M_1$ detrital illite/muscovite (727.6 to 797.9 Ma, cf. Fig. 5), a robust assignment to
provenance areas in the adjacent northern Burdgol zone and Baidrag zone is possible. The
Burdgol zone hosts dominantly metapelites, metapsammites and metacherts, which have an
age of 699 ± 35 Ma, as inferred from K-Ar dating of muscovite (Teraoka et al., 1996), which
closely matches the detrital illite/muscovite ages measured in the sediments from the Valley of
Lakes. The shift towards older ages can be explained by a minor contribution of Neoarchean
rocks from the nearby Baidrag zone (~2.65 Ga old), which are comprised of high-grade
gneisses, charnockites and amphibolites. Both source areas coincide with the heavy mineral
spectra and gravel lithologies of the Valley of Lakes sediments (Höck et al., 1999).



Assuming the detrital illite/muscovite in the Valley of Lakes sediments is a mixture of eroded,
metamorphosed and/or intruded material from both source regions, a relative contribution of
~> 95 % from the Burdgol zone and~< 5 % from the Baidrag zone to the total detrital mica
fraction can be calculated. Detrital silicate influx from the northernmost Bayan Khongor zone,
Dzag zone and Khangai zone is considered to be unlikely, as these source areas are geologically
younger (Ordovician to Cretaceous) (Teraoka et al., 1996). Mixtures of different proportions
of detritus from the Burdgol zone and some younger and older material are unlikely as well, as
constant source proportions over time would be required to explain the same ages for the four
investigated samples. Therefore, the source area relationships of the sediments from the Valley
of Lakes are less complex than previously thought with most detritus delivered from the
regionally adjacent northern areas located within a 100 km range.

5.2 Depositional environment
The poorly sorted, massive to partly cross-bedded sand and gravel beds of the Tsagaan Ovoo
Formation are interpreted as debris flow deposits in alluvial fans, according to the classification
of Miall (1996) for fluvial sediments. These were generated during or soon after heavy rainfall
events, which caused the water-saturated regolith to move down slope (Hubert and Filipov,
1989). The finer, laminated layers with ripple marks, inverse to normal grading and channel
fills deposited in-between the coarser clastic beds represent the background sedimentation in
the upper Eocene, i.e., braided river deposits developed in close vicinity to propagating alluvial
fans (Miall, 1996). Imbrications of pebbles, cobbles and clasts within these beds suggest a
palaeo-current direction from north to south (Höck et al. 1999), which is consistent with major
sediment source areas in the northern Burdgol Zone. Although we found no petrographic-
sedimentological evidence for sediment deposition in a lake or playa environment in the upper
Eocene, as previously proposed by Badamgarav (1993) and Daxner-Höck et al. (2017), the



scatter in the $\delta^{18}O$ isotopic composition of the soil carbonates, which has been attributed to
varying amounts of evaporation (Richoz et al., 2017), may support this assertion.
The poorly sorted, often horizontally bedded and fossiliferous clay-silt-sand(stone) beds of the
Hsanda Gol Formation were deposited in a complex environment: the finer beds have likely
been developed in ephemeral lakes or braided rivers systems draining proximal alluvial fans,
as indicated by the presence of channel sand bodies with basal channel scour lags and cross-
bedded sand fill. The sandier beds are interpreted as open steppe deposits, which have been
temporarily affected by ephemeral river and playa lake sedimentation (Miall, 1996), as it can
be inferred from occasional mud cracks and salt crusts (halite; Höck et al., 1999). On the
contrary, Sun and Windley (2015) have proposed an eolian origin for the Oligocene sediments
and interpreted them as loess deposits, which were transported by westerly winds, based on
REE patterns and comparison with grain size distributions obtained from recent Loess deposits
from Kansas (USA) and the Chinese Loess Plateau. Although we cannot exclude long-distance
transport and subsequent deposition of dust has contributed to at least a minor proportion to
the total basin fill of the Valley of Lakes, we found no petrographic evidence for any aeolian
influences, such as ripples, coarsening up laminae or climbing translatent strata, ventifacts,
mud curls or even quartz grains with crescentic percussion marks (Kenig, 2006; Li et al., 2020).
The lithological variability of the Loh Formation (i.e., poorly sorted and highly fossiliferous
clay-silt-sand-gravel beds deposited in alternate mode) can be best explained by a combination
of debris flow deposits in alluvial fans (coarse clastic material) and abandoned channel deposits
and waning flood sedimentation (fine clastic material) of a shallow, perennial flowing braided
river system Miall (1996). Imbrication of gravels and flow structures in the basalt III group still
indicate a palaeo-current direction from north to south (Höck et al., 1999), which suggests the
Burdgol Zone is the main source area at least up to the upper lower Miocene.



5.3 Origin of hairy illite and Ilt-Smc
Höck et al. (1999) and Sun and Windley (2015) have proposed an aeolian origin or a coupled
aeolian-fluviatile origin for the finest fraction of the Valley of Lakes sediments, while Richoz
et al. (2017) concluded the finest fraction is authigenic and has been formed during or shortly
after the flows of the different basalt groups. However, in none of the above studies radiometric
ages of the clay fraction have been presented to confirm their assertions. Our XRD and SEM
study shows the clay mineral fraction of the Oligocene Hsanda Gol Formation is dominated by
hairy illite and subordinate flake-shaped Ilt-Smc, which cover detrital grains or grow into the
pore space (Fig. 4). All these features that are typical for authigenic illitic phases (Güven et al.,
1980; Rafiei et al., 2020). The polytype analysis and K-Ar age dating reveal these illitic phases
have been precipitated between 34.2 and 25.2 Ma (Fig. 5), which (within uncertainty) is well
within the documented intrusion ages of the basalt I group (32.4-29.1 Ma) and basalt II group
(28.7-24.9 Ma) (Daxner-Höck et al., 2017) and closely matches the biozonation reported in
Harzhauser et al. (2017).
The origin of Ilt-Smc in the Valley of Lakes sediments is difficult to constrain: it could have
been formed during low temperature pedogenesis from smectite or kaolinite precursors of
'zero' age (Huggett et al., 2016), which were deposited due to wind (allochthonous clay source)
or soil water (autochthonous source) action, through a dissolution-(re)precipitation mechanism.
Pedogenic degradation of detrital illitic minerals to produce Ilt-Smc under acidic conditions at
low temperature has also been observed (Meenakshi et al., 2020). Contrary, several published
studies question a low temperature origin of Ilt-Smc in sedimentary successions: Ilt-Smc found
in paleosols from the Illinois Basin was shown to be the alteration product of siliceous parental
phases, which interacted with hydrothermal brines generated during burial diagenesis rather
than of ancient soil formation processes (McIntosh et al., 2020). Środoń (1984) concluded that
smectite and Ilt-Smc phases are relatively stable in surface-near surroundings until the elevated



temperatures of deep diagenesis are reached, which is consistent with slow kinetics of smectite
illitization calculated for shallow buried sediments and/or low temperature settings (Cuadros,
2006). In the case of the Valley of Lakes sediments, the relatively low Ilt content in Ilt-Smc
(~10-30 % Ilt layers) and the stratigraphic age-progression of the authigenic illitic phases up-
section in the sedimentary succession may indicate a pedogenic origin of the Ilt-Smc.
Contrary to the Ilt-Smc, a pedogenic origin of the hairy illite is unlikely, because the formation
of this mineral phase requires temperatures well around 100 °C (Güven et al., 1980; Nadeau et
al. 1985; Baldermann et al., 2017), which is unrealistic high to occur in a developing soil profile
that has experienced a maximum burial depth of only a few hundred meters (Richoz et al.,
2017). The high Ilt content (> 95 % Ilt layers) and the hairy appearance of the illite argue for a
formation at elevated temperatures, which likely developed simultaneously or shortly after the
prominent and recurrent basalt flows, consistent with a basalt-mediated diagenesis. Under such
conditions, pore fluids rich in $K^+$, $Al^{3+}$ and silicic acid are generated through the dissolution of
unstable components, such as feldspar, which subsequently infiltrated the poorly consolidated
(porous) Valley of Lakes sediments, thereby promoting the direct precipitation and growth of
hairy illite in open pores (Fig. 4) and/or the hydrothermal alteration of pre-existing pedogenic
Ilt-Scm to hairy illite (Baldermann et al., 2017). This mechanism is applicable to explain the
shift of the chemical data towards the K pole in the A-CN-K ternary diagram (Fig. 6).

5.5. Palaeo-climate and weathering conditions
Climatic conditions are a primary control of the intensity and type of terrestrial weathering
processes, where humid periods favor chemical weathering and arid periods favor physical
weathering (Chamley, 1989). Analogously, hydroclimatic conditions take a key control on the
intensity of pedogenic processes, which can be recorded in the $\delta^{13}C$ and $\delta^{18}O$ isotopic signature
of authigenic carbonates (i.e., calcrete in paleosols), where wetter conditions favor an excursion





towards lighter $\delta^{13}$C and $\delta^{18}$O values and drier conditions favor an excursion towards heavier
$\delta^{13}$C and $\delta^{18}$O values (Richoz et al., 2017). Hence, variations in chemical weathering indicators
(CIA, PIA and CIW) and in the $\delta^{13}$C and $\delta^{18}$O profiles of soil carbonates across a sedimentary
succession can be used to trace and assess fluctuations in the climatic conditions that prevailed
in the source areas and in the sedimentary basin at the time of sediment deposition, and during
pedogenesis (Nesbitt and Young, 1982; Bahlburg and Dobrzinski, 2011; Fischer-Femal and
Bowen, 2020; Kelson et al., 2020; Zamanian et al., 2021). The formation of soil carbonates is
a highly complex process that can complicate the interpretation of their $\delta^{13}$C and $\delta^{18}$O isotopic
values (Richoz et al., 2017), as global climatic trends may be overprinted by regional factors,
such as contamination with detrital carbonates, dolomitization, meteoric diagenesis, maturation
or oxidation of organic matter, dis-equilibrium conditions between atmospheric (or biogenic)
$CO_2$ and soil solution, evaporation, basalt hydrothermalism, etc. (Kaufman and Knoll, 1995;
Kent-Corson et al., 2009; Caves et al., 2014; Li et al., 2016; Baldermann et al., 2020; Li et al.,
2020). However, if considering that the pristine soil carbonate $\delta^{13}$C and $\delta^{18}$O isotopic signature
is almost well preserved in the Valley of Lakes sediments, their use for palaeo-environmental
reconstructions is possible.
The analysis of the $\delta^{13}$C and $\delta^{18}$O isotopic profiles recorded in the soil carbonates from the
Valley of Lakes (~34-21 Ma) yielded the following palaeo-climatic trends, which are consistent
with inverse shifts seen in the chemical weathering indices (dashed orange lines in Fig. 8), i.e.,
periods with increased precipitation coincide with higher chemical weathering indices and vice
versa. This inverse relation is a robust recorder of changing humid/arid climatic conditions in
an overall arid climate through the Cenozoic in Central Asia, if considering that the source
areas providing the silicate detritus have not changed over time in the investigated sedimentary
succession. Accordingly, during the late Eocene to the earliest Oligocene comparatively humid
to semi-arid climatic conditions prevailed in Central Asia (phase i); biozone A to bottom part



of biozone B; ~34-31 Ma), which is followed by an early Oligocene aridification (phase ii);
bottom part of biozone B; ~31 Ma) and the establishment of more arid climatic conditions
afterwards until the terminal Oligocene (phase iii); upper part of biozone B to biozone C1-D;
~31-23.5 Ma). A shift back towards comparatively humid to semi-arid climatic conditions is
evident in the late Oligocene to earliest Miocene (phase iv); transition between biozones C1-D
and D; ~23.5-23 Ma), which is followed by the establishment of these conditions in the early
Miocene (phase v); biozone D; ~23-21 Ma).
Global cooling events established from $\delta^{13}C$ and $\delta^{18}O$ isotope records of marine deep-sea
sediments (Zachos et al., 2001; Gallagher et al., 2020), such as the Oi-1a/b Glaciation (~34-33
Ma) or the Oligocene Glacial Maximum (~28 Ma) are barely recorded in the soil carbonate
$\delta^{13}C$ and $\delta^{18}O$ isotope profiles. However, they are visible by increases in chemical weathering
indices at exactly these time intervals (blue bars and arrows in Fig. 8) and correspond to
important faunal turnovers (Harzhauser et al., 2016). The early Oligocene aridification (~31
Ma) is seen by an excursion towards heavier isotopic values between ~55 and 60 m in the rock
record, but do not correspond to an important faunal turnover (Harzhauser et al., 2016). On the
contrary, the Oligocene warming event (~25 Ma), marked by an important extinction of the
mammal community, is not seen in the $\delta^{13}C$ and $\delta^{18}O$ isotopic profiles. However, in the interval
from ~87 to 92 m (upper part of biozone C1) an increase of all chemical weathering indices is
evident, which we attribute to strong illitization and local overprinting of the pristine chemical
signature of these sediments. The following Mi-1 Glaciation (~23 Ma) records high chemical
weathering patterns, but shows the expected excursion towards lighter $\delta^{13}C$ and $\delta^{18}O$ isotopic
values.
The reasons for the Cenozoic climate change are hotly debated in the literature, but a strong
decrease in atmospheric $pCO_2$ (Pagani et al., 2011; Anagnostou et al., 2016), major tectonic
events, such as the collision of India with Asia and progressing exhumation of the Himalaya,



as well as re-adjustments in oceanic gateway configurations are widely considered to have
altered the global ocean/atmosphere circulation patterns (Caves Rugenstein and Chamberlain,
2018). This resulted in large-scale shifts in Earth`s climate at this time, which expressed, for
example, in the formation and expansion of the Antarctica ice-sheets and periods of intensified
chemical weathering on land (Zachos et al., 2001, and references therein).

5.6 Hydroclimate and tectonics evolution in Central Asia
The links between the regional tectonic evolution and climate change in Central Asia have been
extensively studied over the past decades. Recently, Caves Rugenstein and Chamberlain (2018)
have concluded Central Asia has received moisture through the mid-latitude westerlies,
maintaining stable semi-arid to arid climatic conditions ever since the early Eocene, based on
the analysis of $\delta^{18}O$ and $\delta^{13}C$ isotope systematics of more than 7700 terrestrial authigenic
carbonate samples from across Asia. On the contrary, southern Tibet, the central Tibetan
Plateau, China and India dominantly received southerly monsoonal moisture, favoring more
humid climatic conditions in these regions compared to Central Asia (Ingalls et al., 2018;
Sandeep et al., 2018). Our data support this viewpoint: consistently higher $\delta^{18}O$ and $\delta^{13}C$ values
measured for the soil carbonates from the Valley of Lakes (Fig. 8), compared to the surrounding
regions, indicate less precipitation and long-term, sustained arid climatic conditions prevailed
in the late Eocene until the early Miocene (Cerling and Quade, 1993; Kent-Corson et al., 2009;
Takeuchi et al., 2010; Caves et al., 2015; Li et al., 2016; Caves Rugenstein and Chamberlain,
2018). An influence of the height and extension of the Tibetan Plateau or the retreat of the
Parathethys on the hydroclimate in Central Asia at this time (An et al., 2001; Zhang et al., 2007)
is barely documented in the sedimentary record of the Valley of Lakes, but cannot be excluded,
which would express in monsoon-dominant environmental pattern and varying amounts of
precipitation (Zhongshi et al., 2007).
The increase in the $\delta^{13}C$ values of the soil carbonates in the Oligocene and the decrease in the
$\delta^{18}O$ values in the terminal Oligocene are ultimately linked to coupled effects arising from the
Cenozoic global cooling and the uplift of the Tian Shan and Altai from the early Neogene
onward, which caused changes in the seasonality and quantity of precipitation (Hendrix et al.,
1994; Macaulay et al., 2016; Hellwig et al., 2017; Wang et al., 2020). The resultant effects on
the fractionation of $\delta^{18}O$ and $\delta^{13}C$ isotopes in soil carbonates are detailed in Caves Rugenstein
and Chamberlain (2018), but are directly related to the development and the establishment of
the Altai rain shadow front. As a consequence, on the leeward side of the Altai, sustained, long-
term drying occurred, which is expressed by systematic changes seen in the isotope profiles
and chemical weathering indices (Fig. 8). This aridification led to a concurrent extension of the
Gobi Desert, causing shifts and turnovers in mammalian and gastropod assemblages observed
in soils of western Mongolia and in the adjacent eastern Valley of Lakes basin at this time
(Neubauer et al., 2013; Harzhauser et al., 2017; Barbolini et al., 2020). We conclude the
climatic and environmental evolution of Central Asia in the Cenozoic was closely coupled to
global climate change, regional tectonic events and adaptions of the circulation pattern of the
westerly winds, transporting less moisture to continental Mongolia, which favored
aridification.

**Acknowledgements**
The authors acknowledge M. Hierz, J. Jernej, S. Perchthold and A. Wolf (Graz University of
Technology) and S. Šimić (Institute for Electron Microscopy and Nanoanalysis and Graz
Centre for Electron Microscopy), who assisted us with the preparation and analysis of the
samples. R. Quezada-Hinojosa is greatly acknowledged for drawing the lithostratigraphic
profile and for providing the oxygen and carbon isotopic data. This research was funded by the
NAWI Graz Geocenter (Graz University of Technology). Field work and sample acquisition



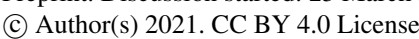



was supported by the Austrian Science Fund (FWF) via P-23061-N19 to G. Daxner-Höck. We
thank our Mongolian and European team members for manifold support during the field work.

**Author contributions**


A.B. wrote the manuscript. W.E.P. carried out field work and collected the samples. O.W. and
S.R. provided the mineralogical and geochemical data. E.A. conducted the discriminant
function analyses. K.W. provided the K/Ar ages. A.B., S.B., S.L., W.E.P. and S.R.
characterized the palaeo-environment and interpreted the stable $\delta^{13}$C and $\delta^{18}$O isotope records.
All authors contributed to the writing of the manuscript.

**Additional information**


Supplementary materials are provided in the electronic appendix to this paper. Requests for
materials and correspondence should be addressed to A.B.

**Competing interests**


All authors declare no competing interests.

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

**Figure Captions / Table Captions**
**Table 1:** Compilation of illite polytype quantification and K-Ar ages of grain size sub-fractions
of sediments collected from (a) TAT section (~90.5 m), (b) TGR-C section (~78.0 m), (c) SHG-
D section (~55.5 m) and (d) TGR-AB section (~35.0 m). The analytical error for the K-Ar age
calculations is given on a 95% confidence level (2σ).

| Sample | Size fraction [μm] | $A_{(-112)}$ [cps·2θ] | 1M [%] | $A_{(114)}$ [cps·2θ] | 2M₁ [%] | 1M_d [%] | $K_2O$ [wt.%] | $^{40}Ar^*$ [nl/g] STP | $^{40}Ar^*$ [%] | Age [Ma] | ± 2SD [Ma] |
|---|---|---|---|---|---|---|---|---|---|---|---|
| TAT | 2-10 | - | - | 0.054 | 21 | 79 | 2.59 | 15.45 | 49.05 | 176.1 | 7.1 |
| TAT | 1-2 | 0.006 | 6 | 0.040 | 16 | 78 | 2.21 | 11.58 | 77.20 | 155.2 | 2.6 |
| TAT | < 1 | 0.012 | 7 | 0.023 | 10 | 83 | 3.39 | 10.75 | 38.18 | 95.8 | 3.2 |
| TGR-C | 2-10 | - | - | 0.038 | 16 | 84 | 2.68 | 13.98 | 81.46 | 155.1 | 2.9 |
| TGR-C | 1-2 | - | - | 0.031 | 13 | 87 | 3.64 | 15.80 | 76.96 | 129.6 | 2.4 |
| TGR-C | < 1 | 0.001 | 5 | 0.027 | 12 | 95 | 3.10 | 12.88 | 66.83 | 124.6 | 1.9 |
| SHG-D | 2-10 | - | - | 0.039 | 16 | 84 | 2.72 | 14.31 | 78.74 | 156.4 | 2.0 |
| SHG-D | 1-2 | - | - | 0.034 | 14 | 86 | 3.86 | 15.93 | 76.09 | 123.6 | 3.2 |
| SHG-D | < 1 | 0.011 | 6 | 0.016 | 8 | 94 | 3.49 | 10.38 | 70.94 | 89.9 | 1.3 |
| TGR-AB | 2-10 | - | - | 0.032 | 14 | 86 | 3.83 | 17.29 | 84.05 | 134.8 | 3.4 |
| TGR-AB | 1-2 | - | - | 0.027 | 12 | 88 | 3.97 | 16.63 | 84.33 | 125.3 | 1.8 |
| TGR-AB | < 1 | 0.032 | 9 | - | 0 | 91 | 0.64 | 0.70 | 10.52 | 33.9 | 3.2 |

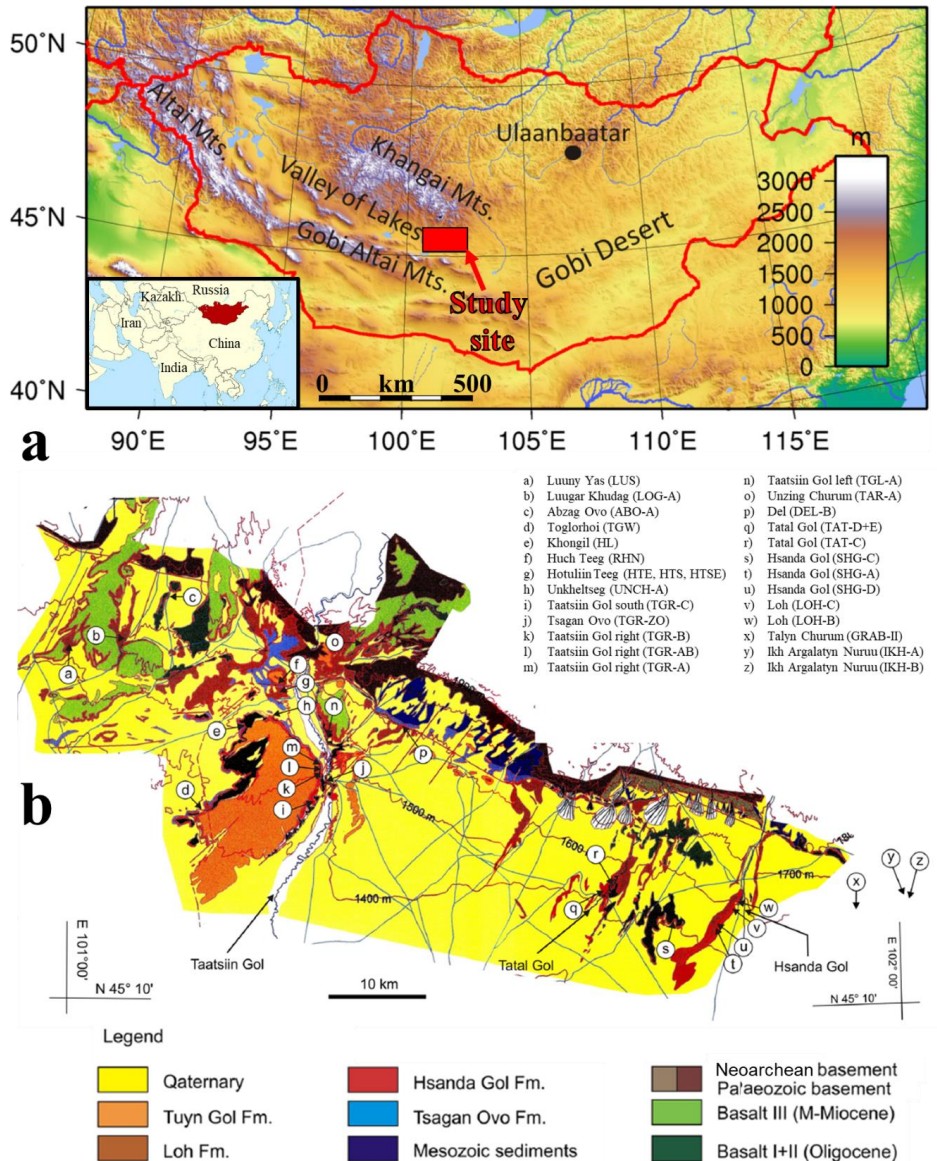

**Figure 1:** (a) Location of the study site in the Taatsiin Gol region, a part of the Valley of Lakes, in Mongolia (Central Asia). Altitude in meters is indicated on the right. (b) Geological map of the Taatsiin Gol area within the Valley of Lakes with the sampling sites marked in alphabetical order (modified after Daxner-Höck et al., 2017).



906

**Figure 2:** (a) Integrated lithostratigraphic profile of the investigated sedimentary succession

from the Taatsiin Gol region, Valley of Lakes (modified after Richoz et al., 2017), with

biozonation (modified after Harzhauser et al., 2017). (b-d) Field impressions of the sections

Hotuliin Teeg (HTE) with calichized basalt II group, Tatal Gol (TAT-E) sediments and Taatsiin

Gol right (TGR-B) section with basalt I group (modified after Daxner-Höck et al., 2017).

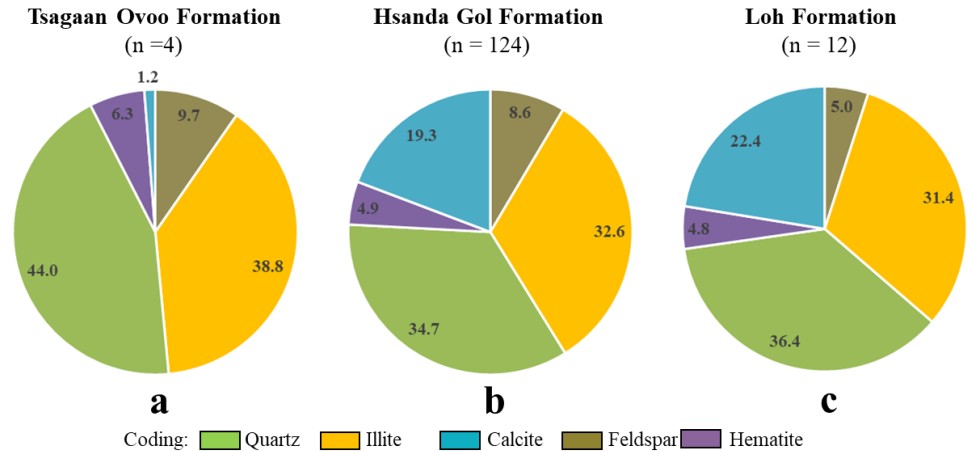

912

**Figure 3:** Averaged mineralogical composition (in wt%) of the sediments from the (a) upper

Eocene Tsagaan Ovoo Formation, (b) Oligocene Hsanda Gol Formation and (c) lower Miocene

Loh Formation from the Valley of Lakes, determined by XRD analysis.

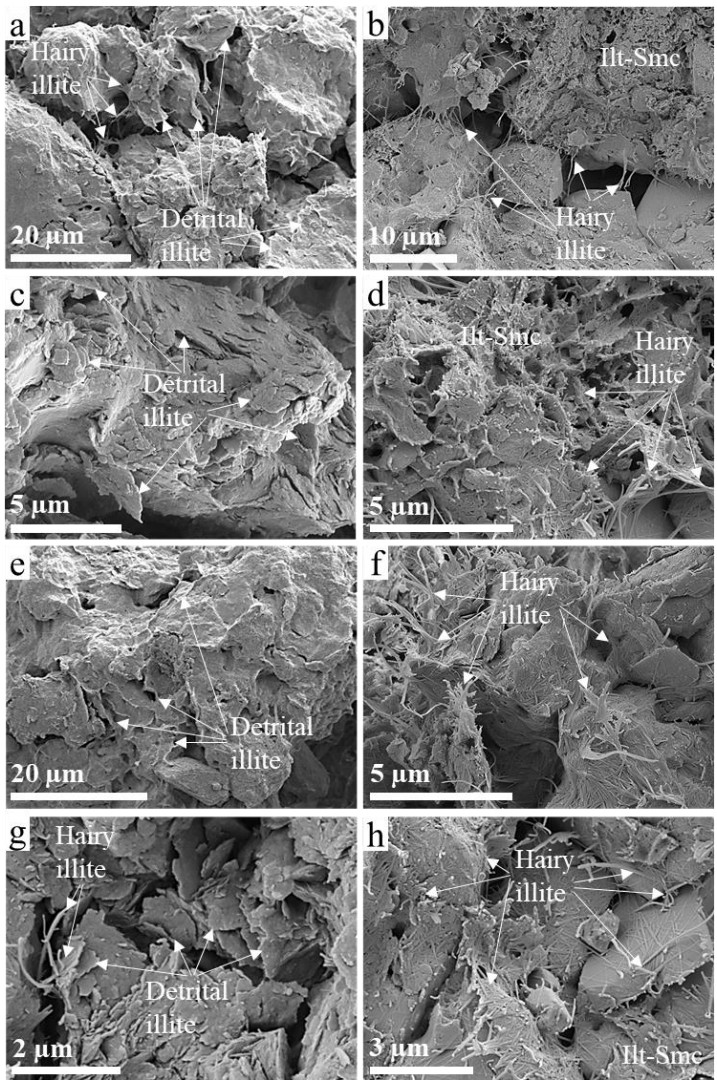

916

**Figure 4:** Secondary electron images of partly calichized and illitized silty to sandy deposits

from the of the Oligocene Hsanda Gol Formation, Valley of Lakes, collected from (a-b) TAT

section (~90.5 m), (c-d) TGR-C section (~78.0 m), (e-f) SHG-D section (~55.5 m) and (g-h)

TGR-AB section (~35.0 m). The detrital illite/muscovite (left panel) occurs as coarse, rounded

or pseudohexagonal platelets, whereas authigenic illite-smectite (Ilt-Smc) and hairy illite (right

panel) appear either as fine, flaky to irregular particles or as long, but thin laths and fibers, both

covering detrital grains or growing into the open pores.



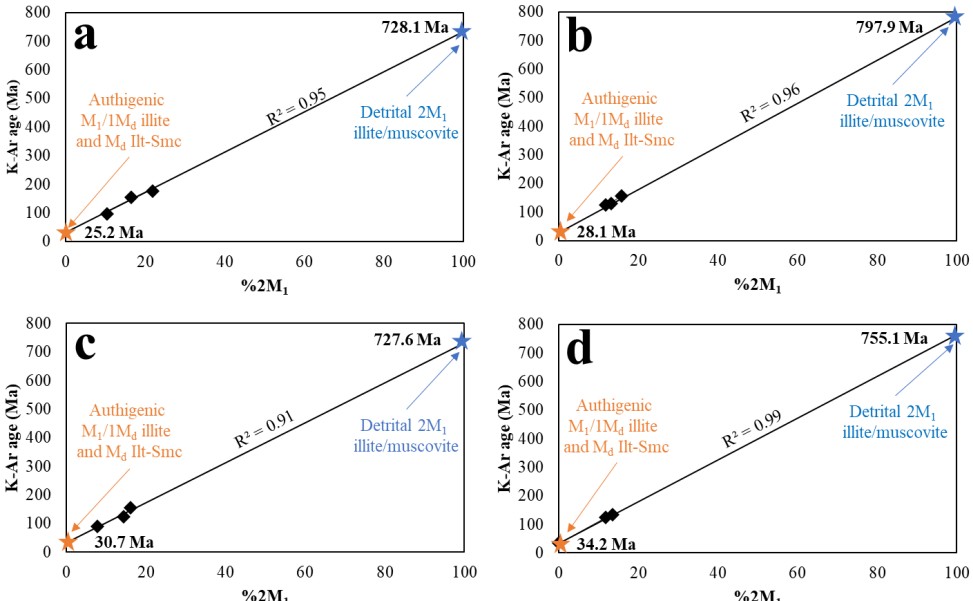

924

**Figure 5:** Crystallization ages of detrital $2M_1$ illite/muscovite and of authigenic $1M_d$/$1M$ illite

and illite-smectite (Ilt-Smc) from the Valley of Lakes, calculated for sediments collected from

(a) TAT section (~90.5 m), (b) TGR-C section (~78.0 m), (c) SHG-D section (~55.5 m) and

(d) TGR-AB section (~35.0 m) using illite polytype quantification and K-Ar age systematics

of different grain size sub-fractions (from left to right: < 1 µm, 1-2 µm and 2-10 µm).

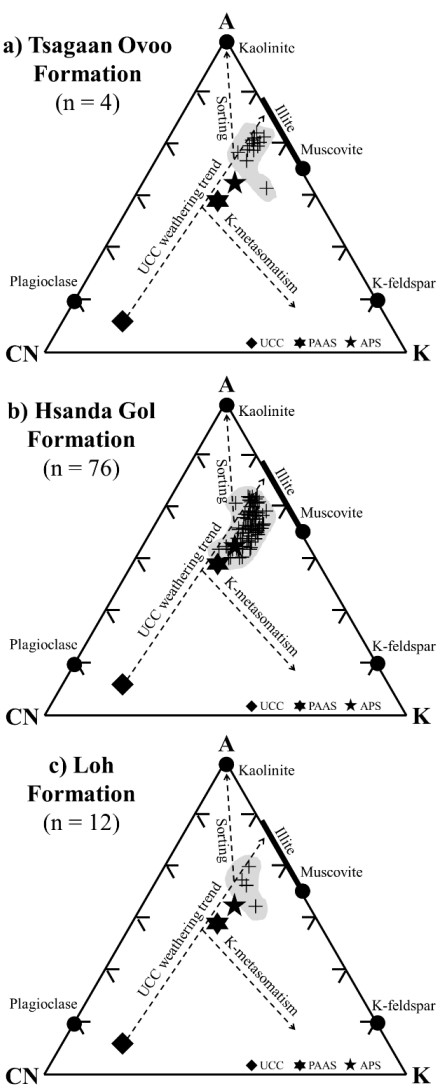

930

**Figure 6:** $Al_2O_3$-$(CaO*+Na_2O)$-$K_2O$ (A-CN-K) ternary diagram of Nesbitt and Young (1984)

showing the compositional ranges of sediments from the Valley of Lakes from (a) upper

Eocene Tsagaan Ovoo Formation, (b) Oligocene Hsanda Gol Formation and (c) lower Miocene

Loh Formation. Note that most samples are shifted to the K pole of the diagram, which indicates

a post-depositional enrichment of $K_2O$ due to illitization. The composition of Upper

Continental Crust (UCC), Average Proterozoic Shale (APS) and Post-Archean Australian

Shale (PAAS) are included for comparison.



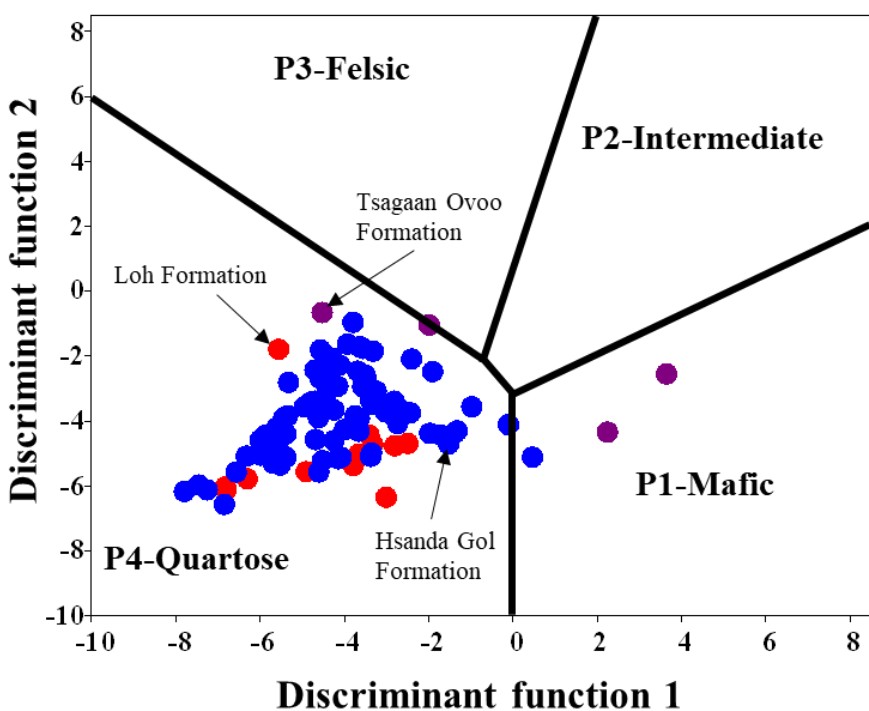

938

**Figure 7:** Discrimination plot of discriminant function 1 and 2 indicating a narrow provenance

range (mainly type P4-quartzose) for the sediments from the Valley of Lakes, Mongolia.



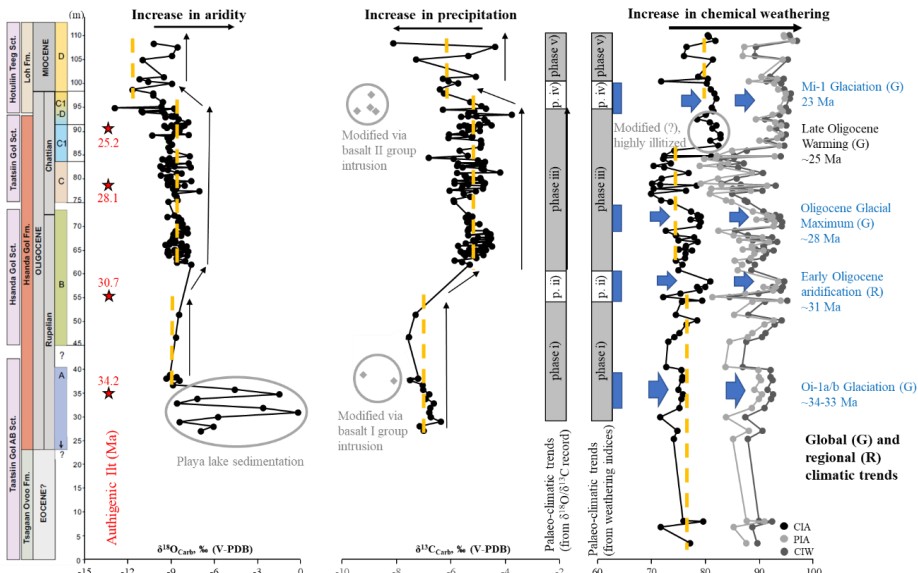

941

**Figure 8:** Lithostratigraphic framework of the sediments from the Valley of Lakes (Mongolia,

Central Asia) showing the biozonation (modified after Harzhauser et al., 2017) and formation

ages of authigenic illitic (Ilt) phases obtained in this study (red asterisks), as well as soil

carbonate $\delta^{18}$O and $\delta^{13}$C isotope profiles and shifts in the silicate mineral-derived chemical

weathering indicators. Note that these hydroclimate proxies are inversely correlated and follow

long-term trends (indicated by orange dashed lines) in aridification or gain of humidity in this

region (indicated by black arrows). Increased chemical weathering degrees (highlighted with

blue bars and blue arrows) coincide with glaciation events documented in time-equivalent

marine deep-sea deposits (Zachos et al., 2001; Gallagher et al., 2020). Samples and intervals

outlined with grey circles are most likely modified due to the flows of the basalt I and II groups

or local strong illitization, and are therefore excluded from the palaeo-climatic interpretation.