# Peer review of "Palaeo-environmental evolution of Central Asia during the Cenozoic: New insights from"

_Climate of the Past, 2021_

## Author Response (AR1)

**Review of paper Preprint cp-2021-32: "Palaeo-environmental evolution of Central Asia during the Cenozoic: New insights from the continental sedimentary archive of the Valley of Lakes (Mongolia)" by Baldermann et al.**

**Editor Decision: Reconsider after major revisions (13 Jul 2021) by Zhengtang Guo**

Dear Andre Baldermann,

Thank you for submitting your work to Climate of the Past, and for your pre-responses to referees' comments. As you had already examined, both referees indicated the significance of the new data. However, they both raised some issues that would require your considerations through a major revision. I am looking forward to receiving your revised manuscript.

With the best wishes,

Zhengtang GUO

We thank the handling editor, Dr. Zhengtang Guo, as well as the two referees for their helpful and insightful comments, which we have all considered in the revised manuscript. In the below text, we explain how we have revised our manuscript based on the criticism we received from the reviewers. We have further revised our references in accordance with the CP style.

**RC1: Anonymous Reviewer**

**General comments**

The valley of lakes in Mongolia is certainly a key area for investigating Cenozoic mammal evolution and climate changes in Central Asia. It is significance to reconstruct the paleoclimate evolution history during late Eocene to early Miocene based on sedimentological, petrographic, mineralogical and geochemical signatures recorded in a sedimentary succession in the valley of lakes in Mongolia. In this study, Baldermann et al. extended the existing mineralogical and (isotope) geochemical dataset reported in Richoze et al. (2017) to constrain provenance, paleoevironmental conditions and post-depositional alteration history of the Eocene-Miocene sedimentary succession. Their reconstruction provides good data support for refining the evolution of hydroclimate and weathering conditions in Central Asia in the early Cenozoic. However, there are still some main issues that need further discussion.

We thank the reviewer for the overall positive evaluation of our work. Below, we comment on the specific comments provided by the reviewer and indicate how we have revised the text of our manuscript accordingly.

**Specific comments**

1) The chronological framework for sedimentary succession is the basis of paleoclimate reconstruction. In this study, authors thought that authigenic "hairy" illite minerals were formed during coupled petrogenesis and precipitation from hydrothermal fluids originating from major basalt flow events, and illite crystallization ages in sedimentary succession were used to establish the chronological framework in this study. Noticeable, the age of basalt I is ~31.5 Ma at ~40-45m (as shown in Figure 2), which is much younger than illite crystallization age (34.2 Ma) at ~35 m. Authigenic illite crystallization ages possibly are ages when sedimentary strata were affected by hydrothermal fluids, should not be the ages when the sedimentary strata were deposited. Therefore, it should be careful to use the illite crystallization ages to establish the chronological framework of sedimentary succession. Detailed magnetostratigraphic work in the valley of lakes in Mongolia had been done by Sun and Windley (2015). It is suggested to consider their established magnetostratigraphic age framework in this study.

We fully agree with the reviewer. Illitization post-dates the deposition of the sedimentary strata of the Valley of Lakes, and was likely associated with pedogenesis and the major basalt flow events. We state this in section 5.3: "The polytype analysis and K-Ar age dating reveal these illitic phases have been precipitated between 34.2 and 25.2 Ma (Fig. 5), which (within uncertainty) is well within the documented intrusion ages of the basalt I group (32.4-29.1 Ma) and basalt II group (28.7-24.9 Ma) (Daxner-Höck et al., 2017) and closely matches the biozonation reported in Harzhauser et al. (2017)." We therefore agree with the reviewer that the lowermost illite age (34.2 Ma) is slightly younger than the intrusion ages of the basalt I group (32.4-29.1 Ma), but still within the analytical uncertainty of K/Ar age dating. We have changed the above sentence as follows: "The polytype analysis and K-Ar age dating reveal these illitic phases have been precipitated between 34.2 and 25.2 Ma (Fig. 5), which (within uncertainty of the K-Ar age dating method we have used here) is well within the documented intrusion ages of the basalt I group (32.4-29.1 Ma) and basalt II group (28.7-24.9 Ma) (Daxner-Höck et al., 2017) and closely matches the biozonation reported in Harzhauser et al. (2017)." The biozonation of Harzhauser et al. (2017) we use here for our chronological framework is based on the radiometric and magnetostratigraphic dating of the sections by Höck et al. (1999) and Sun and Windley (2015). Harzhauser et al. (2017) explicitly state in their Introduction: "The radiometric and magnetostratigraphic dating of the sections by Höck et al. (1999) and Sun and Windley (2015) suggests an early Rupelian age for Zone A (33.9 Ma to ~31.5 Ma), a late Rupelian age for Zone B (~31.5 Ma to ~28.1 Ma), a nearly Chattian age for Zone C (~28.1 Ma to ~25.6 Ma), a mid-Chattian age for Zone C1 (~25.6 Ma to ~24.0 Ma), a latest Chattian age for Zone C1-D (~24.0 Ma to ~23.0 Ma) and an Aquitanian age for Zone D (~23.0 Ma to ~21.0 Ma)." As our chronological framework is based on the biozonation of Harzhauser et al. (2017), the magnetostratigraphic work of Sun and Windley (2015) is directly accounted for. For clarification, we have added the precise boundaries of the biozones A to D in the geological framework section (section 2, second last paragraph) and have also provided these boundaries in Figure 8, together with the illite formation ages. In summary, the global and regional climatic trends seen in the Valley of Lakes sediments (Figure 8) are supported by a well-established chronological framework.

2) As mentioned in this paper, the depositional setting was characterized by an ephemeral braided river system draining prograding alluvial fans, with episodes of lake, playa or open steppe sedimentation. It means that the sedimentary facies in the study area have been changed many times during late Eocene to early Miocene. The chemical weathering index may change with different sedimentary facies. Therefore, it is suggested that sedimentary facies should be added to the Figure 8.

In section 2, we refer to published literature that addresses in detail the changes observed in the sedimentary facies across the different sections of the Valley of Lakes: "Further details about the local nomenclature, the investigated profiles, profile correlation and lithostratigraphic relationships are provided in Harzhauser et al. (2017), Daxner-Höck et al. (2017) and Richoz et al. (2017)." We don`t find it necessary to repeat these findings here. Nevertheless, Richoz et al. (2017) have concluded that the overall sedimentation system has not changed much in the considered timeframe, a feature confirmed in this study. We state this now explicitly in section 5.1. Moreover, our novel K-Ar datings of the detrital illite fraction as well as our discrimination function analysis indicate no significant changes in sediment provenance occurred from the late Eocene to the early Miocene. Alike, we propose an about constant detrital silicate influx with a relative contribution of ~> 95 % from the Burdgol zone and~< 5 % from the Baidrag zone. We therefore conclude (end of section 5.1): "Thus, variation in the chemical weathering indices outlined below most likely record changes in the weathering conditions of the source rock areas rather than changes in the sedimentary facies at the same time."

3) The scatter in the δ18O isotope composition of the soil carbonates in the upper Eocene was attributed to playa lake sedimentation (as shown in Figure 8), but there was no petrographic-sedimentological evidence for sediment deposition in a lake or playa environment. Why is there such a paradox?

We have changed the sentence as follows for clarification: "In contrast to Badamgarav (1993) and Daxner-Höck et al. (2017), we found no petrographic-sedimentological evidence for lake or playa sedimentation in the upper Eocene strata, which we attribute to the different sample types considered: While Badamgarav (1993) and Daxner-Höck et al. (2017) identified efflorescent salt crusts composed of halite, tepees and polygonal structures in some sedimentary layers, no such structures were observed in the paleosol horizons of the same age. However, the scatter in the $\delta^{18}O$ isotopic composition of the soil carbonates, which has been attributed to varying amounts of evaporation (Richoz et al., 2017), is consistent with a playa lake setting."

4) The δ13C and δ18O profiles showed that significant aridification occurred between ~62-92 m (maybe ~30-24 Ma) in the valley of lakes, and the aridity weakened above ~95 m (after ~24 Ma). The change trend in chemical weathering indexes were not consistent with δ13C and δ18O profiles. In the range of 50-85m (maybe ~31-26 Ma), chemical weathering indexes fluctuated frequently, but generally decreased; they increased significantly at ~26 Ma, and maintained relatively stable high values during the early Miocene. What causes the difference between isotope data and chemical weathering indexes? Sedimentary facies? Post diagenesis? Basalt flow events? Or reginal tectonic activities? Noticeable, without the precise chronological framework, it is not significant to make one-to-one correspondence between the fluctuations of chemical weathering indexes and global climate events.

As indicated in our response to comment 1) we are confident that the chronological framework we use is correct. We agree with the reviewer that the weathering indices scatter to some degree but they are basically inversely correlated to the $\delta^{13}$C and $\delta^{18}$O profiles (cf. dashed orange lines in Fig. 8). This is because variations in the $\delta^{13}$C and $\delta^{18}$O profiles are consistent "with inverse shifts seen in the chemical weathering indices (dashed orange lines in Fig. 8), i.e., periods with increased precipitation coincide with higher chemical weathering indices and vice versa." Thus, the palaeo-climatic conditions in the Valley of Lakes and in the adjacent areas were the driving factor for the observed hydroclimate and weathering trends. Changes in sedimentary facies, diagenesis, basalt flow events or reginal tectonic activities are negligible as the trends we see are based on a stable sediment provenance and pristine soil carbonate isotope signals. We have added a statement in the second paragraph of section 5.5. stating this.

**Technical corrections**

1) The formation names marked in Figure 6 are wrong, please check it carefully. e.g. a) Tsagaan Ovoo formation should be Loh Formation. c) Loh should be Tsagaan

The formation names marked in Figure 6 are correct but we have changed sub-figures a) and c) in order to bring the formations in stratigraphic order.

Sun, J.M.& Windley, B.F. (2015). Onset of aridification by 34 Ma across the Eocene-Oligecene transition in Central Asia. Geology, 43(11), 1015-1018.

**RC2: Jeremy Caves Rugenstein**

Baldermann and co-authors provide new data from the well-studied Valley of Lakes section in central-southern Mongolia to understand the sedimentological and paleo-environments during late Paleogene and early Neogene Mongolia. The authors find that a number of paleo-environmental indicators, such as CIA, track global climate signals, but that d18O and d13C do not; they conclude that stable isotopes of authigenic carbonates in this section reflect, to a much greater extent, uplift of the Altai and Tian Shan.

I found this paper easy to read; the figures support the text, and; the paper is well-referenced. I believe this paper is appropriate for a journal such as Climate of the Past subject to minor revisions. Below, I present a few comments, which I think will make the paper more robust. Please note that I am not an expert on Ar-dating of clays; I therefore restrict my comments to the paleo-environmental aspects of the paper.

We thank the reviewer for the very positive evaluation of our work. Below, we comment on the specific comments provided by the reviewer and indicate how we have revised the text of the manuscript accordingly.

I'm curious why the stable isotopes—particularly the d13C—do not track with the weathering indices, such as CIA. The authors interpret their d13C record in terms of precipitation; strictly, this isn't correct particularly over long timescales. Rather, d13C records the balance between atmospheric CO2 and the soil respiration flux (Cerling, 1999, 1984; Cerling and Quade, 1993). Over this time frame, changes in atmospheric CO2 need to be considered. However, for most of Asia, changes in plant productivity—probably driven by changes in the atmospheric CO2 via the CO2 fertilization effect—seem to be the larger driver of soil carbonate d13C changes (Caves et al., 2016; Caves Rugenstein and Chamberlain, 2018). This is likely to have an effect on weathering, since plant-produced CO2 plays a vital role in breaking down primary minerals. Thus, it is curious why these weathering indices and d13C are decoupled, and some speculation from the authors on why would be helpful. We recently published a paper that dealt with this issue in the late Cretaceous Songliao Basin in NE China (Gao et al., 2021).

We fully agree but want to note here that Richoz et al. (2017) have commented on this issue: "From ~33 to 22 Ma, the atmospheric $CO_2$ concentration decreased from 800 to 200 ppm (Zhang et al. 2013), which should be translated in a trend towards lighter $\delta^{13}C$ soil values. We do not see this trend in our data, and thus, changes in aridification in Central Mongolia may have overprinted this effect." We have added the following explanation to the text (end of second paragraph, section 5.5): "We note here that the atmospheric $CO_2$ concentration decreased from 800 ppm to 200 ppm from ~33 to 22 Ma (Zhang et al. 2013), which should have shifted the soil carbonate $\delta^{13}C$ signatures towards lighter values. However, due to changes in aridification in Central Mongolia at the same time, this trend is not seen in the data. Indeed, an increase in aridification results in a restricted soil moisture content that can i) increase the $\delta^{13}C$ value of soil carbonates, ii) causes the plant productivity to decrease, which affects the ratio of atmospheric $CO_2$ to soil respired $CO_2$ and iii) reduce the formation depth of the soil carbonates and thus the relative contributions of atmospheric $CO_2$ and soil-derived carbon (Cerling and Quade 1993; Caves et al. 2014). As a consequence, the $\delta^{13}C$ isotopic signature of the soil carbonate is linked to aridification pulses, which also affects the weathering intensity of the sediment source areas, explaining the inverse relation between the isotope record and the chemical alteration indices."

The relative lack of change in d18O is not too surprising. In such a continental, semi-arid setting as the Valley of Lakes, small changes in hydroclimate are unlikely to produce changes in d18O, given that most moisture is recycled in this setting and there is very little runoff. Such predictions for meteoric water d18O in continental settings has been detailed in a number of studies (Caves et al., 2015; Chamberlain et al., 2014; Kukla et al., 2019; Winnick et al., 2014).

We thank the reviewer for this excellent explanation and have added the following sentence after the aforementioned insertion: "On the contrary, large changes in the $\delta^{18}O$ isotopic record of pristine soil carbonates are not to be expected given that the hydroclimatic variations are small in the semi-arid setting of the Valley of Lakes and that most moisture is recycled (Caves et al., 2015; Chamberlain et al., 2014; Kukla et al., 2019; Winnick et al., 2014)".

I'm curious why the authors attributed many of the paleo-environmental changes to uplift of the Tian Shan and Altai mountains, rather than uplift of the Hangay mountains to the north. There is, of course, some dispute about the paleo-elevation of the Hangay mountains through time (McDannell et al., 2018; Sahagian et al., 2016) and my own work (Caves et al., 2014) suggests that the Hangay play an important role in blocking moisture to this part of the Valley of Lakes. Some discussion of why the authors have decided to attribute hydroclimatic changes to uplift of the Tian Shan and Altai versus changes in Hangay paleo-elevation would be appropriate and would be of interest to a broad swath of researchers who are interested in tectonics and paleoclimate in Mongolia.

We fully agree with the reviewer. We have added the following explanation to section 5.6):

"Moreover, the progressive uplifting of the Hangay mountains to the north ever since the early

Oligocene also blocked Siberian moisture transport to the northern Gobi, as it can be inferred from $\delta^{13}C$ and $\delta^{18}O$ isotope signatures recorded in paleosol carbonates from different transects at the northern edge of the Gobi Desert and in the lee of the Altai and Hangay mountains, and consequently contributed to the aridification of this area (Caves et al., 2014; Sahagian et al.,

2016; McDannell et al., 2018)."

Minor Comments:

Line 90: I think you mean to cite Xiao et al., 2010 here.

We have changed the reference accordingly.

Figure 8: How is the position of the dashed yellow, vertical lines in the d18O panel positioned?

For the uppermost samples, is this line placed along the minimum values because there is evidence that there is evaporative effects for the higher d18O samples? What evidence is this?

The dashed yellow, vertical lines represent the moving average. We have moved the line to the right of the $\delta^{18}O$ isotope record, thank you for this comment. In addition, we have added the biozone ages for clarification.

Caves, J.K., Moragne, D.Y., Ibarra, D.E., Bayshashov, B.U., Gao, Y., Jones, M.M.,

Zhamangara, A., Arzhannikova, A. V., Arzhannikov, S.G., Chamberlain, C.P., 2016. The

Neogene de-greening of Central Asia. Geology 44, 887–890. https://doi.org/10.1130/G38267.1

Caves, J.K., Sjostrom, D.J., Mix, H.T., Winnick, M.J., Chamberlain, C.P., 2014. Aridification of Central Asia and Uplift of the Altai and Hangay Mountains, Mongolia: Stable Isotope

Evidence. Am. J. Sci. 314, 1171–1201. https://doi.org/10.2475/08.2014.01

Caves, J.K., Winnick, M.J., Graham, S.A., Sjostrom, D.J., Mulch, A., Chamberlain, C.P., 2015.
Role of the westerlies in Central Asia climate over the Cenozoic. Earth Planet. Sci. Lett. 428,
33–43. https://doi.org/10.1016/j.epsl.2015.07.023

Caves Rugenstein, J.K., Chamberlain, C.P., 2018. The evolution of hydroclimate in Asia over
the Cenozoic: A stable-isotope perspective. Earth-Science Rev. 185, 1129–1156.
https://doi.org/10.1016/j.earscirev.2018.09.003

Cerling, T.E., 1999. Stable carbon isotopes in palaeosol carbonates, in: Thiry, M., Simon-
Coincon, R. (Eds.), Palaeoweathering, Palaeosurfaces and Related Continental Deposits. The
International Association of Sedimentologists, pp. 43–60.
https://doi.org/10.1002/9781444304190.ch2

Cerling, T.E., 1984. The stable isotopic composition of modern soil carbonate and its
relationship to climate. Earth Planet. Sci. Lett. 71, 229–240. https://doi.org/10.1016/0012-
821X(84)90089-X

Cerling, T.E., Quade, J., 1993. Stable Carbon and Oxygen Isotopes in Soil Carbonates, in:
Swart, P., Lohmann, K., McKenzie, J., Savin, S. (Eds.), Climate Change in Continental Isotopic
Records. American Geophysical Union, Washington, DC, pp. 217–231.
https://doi.org/10.1029/GM078p0217

Chamberlain, C.P., Winnick, M.J., Mix, H.T., Chamberlain, S.D., Maher, K., 2014. The impact
of Neogene grassland expansion and aridification on the isotopic composition of continental
precipitation. Global Biogeochem. Cycles 28, 1–13.
https://doi.org/10.1002/2014GB004822.Received

Gao, Yuan, Ibarra, D.E., Rugenstein, J.K.C., Kukla, T., Methner, K., Gao, Youfeng, Huang,
H., Lin, Z., Zhang, L., Xi, D., Wu, H., Carroll, R., Graham, S.A., Chamberlain, C.P., 2021.
Terrestrial climate in mid-latitude East Asia from the latest Cretaceous to the earliest
Paleogene: A multiproxy record from the Songliao Basin in northeastern China. Earth-Science
Rev. 103572. https://doi.org/10.1016/j.earscirev.2021.103572

Kukla, T., Winnick, M.J., Maher, K., Ibarra, D.E., Chamberlain, C.P., 2019. The Sensitivity of
Terrestrial δ18O Gradients to Hydroclimate Evolution. J. Geophys. Res. Atmos. 124, 563–582.
https://doi.org/10.1029/2018JD029571

McDannell, K.T., Zeitler, P.K., Idleman, B.D., 2018. Relict Topography Within the Hangay

Mountains in Central Mongolia: Quantifying Long-Term Exhumation and Relief Change in an

Old Landscape. Tectonics 37, 2531–2558. https://doi.org/10.1029/2017TC004682

Sahagian, D., Proussevitch, A., Ancuta, L.D., Idleman, B.D., Zeitler, P.K., 2016. Uplift of

Central   Mongolia   Recorded   in   Vesicular   Basalts.   J.   Geol.   124,   435–445.

https://doi.org/10.1086/686272

Winnick, M.J., Chamberlain, C.P., Caves, J.K., Welker, J.M., 2014. Quantifying the isotopic

"continental   effect."   Earth   Planet.   Sci.   Lett.   406,   123–133.

https://doi.org/10.1016/j.epsl.2014.09.005

---

## Author Response (AR2)

**Review of paper Preprint cp-2021-32: "Palaeo-environmental evolution of Central Asia during the Cenozoic: New insights from the continental sedimentary archive of the Valley of Lakes (Mongolia)" by Baldermann et al.**

In the below text we comment on all of the reviewer comments and outline our point by point responses, demonstrating our ability to address all substantial comments.

**RC1: Anonymous Reviewer**

Some references should be revised in accordance with the CP style. For example, Lines 85 and 946,"Zhongshi" should be "Zhang".

We thank the reviewer for this note and have corrected the reference accordingly.

Best regards,

Andre Baldermann